# Creating Majorana modes from segmented Fermi surface

Michał Papaj [1✉] & Liang Fu[1✉]

Majorana bound states provide a fertile ground for both investigation of fundamental phenomena as well as for applications in quantum computation. However, despite enormous experimental and theoretical efforts, the currently available Majorana platforms suffer from a multitude of issues that prevent full realization of their potential. Therefore, improved Majorana systems are still highly sought after. Here we present a platform for creating Majorana bound states from 2D gapless superconducting state in spin-helical systems under the in-plane magnetic or Zeeman field. Topological 1D channels are formed by quantum confinement of quasiparticles via Andreev reflection from the surrounding fully gapped superconducting region. Our proposal can be realized using narrow strips of magnetic insulators on top of proximitized 3D topological insulators. This setup has key advantages that include: small required fields, no necessity of fine-tuning of chemical potential, removal of the low-energy detrimental states, and large attainable topological gap.

---

[1] Department of Physics, Massachusetts Institute of Technology, Cambridge, MA 02139, USA. ✉email: mpapaj@mit.edu; liangfu@mit.edu

Majorana bound states[1,2] have engrossed the condensed matter physics community for much of the last decade, offering promises of fascinating phenomena of fundamental interest and potential applications in topologically protected quantum computing[3,4]. Over the years, multiple platforms of very diverse variety have been proposed as hosts of Majorana bound states and intensively studied, including proximitized topological insulators[5], heavy metal surfaces[6], semiconductor nanowires[7–9], magnetic atom chains[10,11], planar Josephson junctions in 2D electron gas[12] and iron superconductors[13–15]. Signatures of Majorana bound states in all these platforms have been observed in multiple experiments[16–29].

Despite the remarkable progress on many frontiers, most of the existing material platforms have some disadvantages that hamper their further investigation and prompt the continued search[30–32] for new systems that would resolve these issues. The multitude of issues plaguing current Majorana platforms are related to the material quality, the device fabrication process and the required experimental conditions. For example, in iron superconductor FeTe$_{0.55}$Se$_{0.45}$ the topological band inversion necessary for creating Majorana bound states requires alloying, which results in disorder and inhomogeneity[20]. In the fabrication of semiconductor nanowires band bending near the crystal-vacuum interface may result in quasi-Majorana bound states that complicate the interpretation of the zero bias peak[33,34]. Moreover, in many platforms the appearance of Majorana bound states relies on strong external magnetic fields above 1T (which is detrimental to superconductivity) or fine-tuning the chemical potential into the Zeeman gap.

In this work we propose a new approach to creating Majorana bound states in 2D systems that may help to resolve some of these issues. Our proposal is based on a gapless superconducting state of spin-helical electrons placed under either external magnetic field or influence of a magnetic insulator. Examples of such systems are presented in Fig. 1 and include 3D topological insulators (TI) in proximity to conventional superconductors[35–42] or superconductors with Rashba spin-orbit coupling. In such systems the interplay of superconductivity and magnetic field leads to a "segmented Fermi surface": while a large part of the normal state Fermi surface is still gapped, its remainder is reconstructed into contours consisting of electron- and hole-dominated arcs[43]. A topological gap can be opened in a quantum confined quasi-1D channel of such a gapless superconducting state surrounded by regions with a full superconducting gap. Majorana bound states will emerge at the boundaries of thus formed 1D channel.

As a concrete example, we focus on the setup of Fig. 1a, where we propose to use narrow strips of magnetic insulator such as EuS on top of a proximitized thin film of 3D TI such as Bi$_2$Se$_3$[44]. Crucially, since the narrow strip region is surrounded by a superconducting (and not just insulating) gap, the number of low-energy modes depends on the strip width $W$ and the superconducting coherence length $\xi$, independent of the chemical potential. As the Zeeman field is induced by exchange interaction with a magnetic insulator strip, it does not impact the parent superconductor or destroy the proximity effect in the rest of the surface. Since our proposal relies on segmented (rather than full) Fermi surface, it results in the removal of the low-energy states. This translates to large topological gaps that are crucial for topological quantum computing applications. Combination of all these features makes our proposal an attractive alternative to the existing systems.

## Results

**Model**. While our proposal based on segmented Fermi surfaces is rather versatile, for concreteness we start our analysis with a thin film of a 3D topological insulator (TI) in proximity to an ordinary s-wave superconductor[5], with a narrow strip of magnetic insulator deposited on top as shown in Fig. 1a. The TI surface states acquire the superconducting gap $\Delta$ everywhere on the surface and are subject to an exchange field only in the region beneath the magnetic insulator. This system is described by the following Bogoliubov–de Gennes Hamiltonian:

$$H = v(k_x \tau_z s_y - k_y \tau_z s_x) - \mu \tau_z + B_x(x,y)s_x + \Delta \tau_x, \quad (1)$$

where $\tau_i$ and $s_i$ are Pauli matrices describing the particle-hole and spin degrees of freedom, respectively, $v$ is the Fermi velocity of the surface state and $\mu$ is the chemical potential. In our discussion we focus on the case of exchange field parallel to the strip, which results in the Zeeman energy $B_x$.

We first want to analyze the eigenstates of this Hamiltonian for translationally invariant cases and then construct the solutions that are quantum confined within the finite width strip. To simplify the analysis of the problem we note that in the experimentally relevant scenarios $\mu$ is the largest energy scale of the problem ($\mu \gg B_x, \Delta$) and so we can concentrate only on the upper Dirac cone near the Fermi level and disregard the other band (assuming $\mu > 0$). After the projection (as discussed in Supplementary Note 1) we obtain the following low-energy Hamiltonian:

$$H_p = \begin{pmatrix} kv - \mu - B_x k_y/k & \Delta \\ \Delta & -kv + \mu - B_x k_y/k \end{pmatrix} \quad (2)$$

where $k = \sqrt{k_x^2 + k_y^2}$ and we have neglected the off-diagonal momentum dependent terms as they are suppressed by the factors of $\Delta/\mu$, $B_x/\mu$.

This effective Hamiltonian clearly demonstrates that due to the spin-momentum locking of Dirac surface states the Zeeman field $B_x$ causes a Fermi surface shift in the $k_y$ direction. This results in a direction-dependent depairing effect on the superconducting state, which is maximum at $k_y = \pm k_F$ and zero for $k_x = \pm k_F$, with Fermi momentum $k_F = \mu/v$. The eigenvalues of Hamiltonian (2) at $k_x = 0$ are presented in Fig. 2a for two values of magnetic field, $B_x = 0$ and $B_x = 1.5\Delta$. While in the absence of Zeeman energy there are no states inside of the gap, when $B_x > \Delta$ the gap closes and zero energy states of Bogoliubov quasiparticles are located in two closed contours near $k_y = \pm k_F$. Each contour is composed of electron- and hole-dominated arcs, while the remainder of the initial normal state Fermi surface is still gapped around $k_x = \pm k_F$[43]. Such a segmented Fermi surface is presented in Fig. 2(b). The contour at $E = 0$ can be described by the $k_y$ wavevector components parametrized by $k_x$, which are approximately given by:

$$k_{e/h,\pm} = \pm k_0 + s\sqrt{\frac{B_x^2}{v^2} - \frac{\Delta^2}{v^2}\frac{k_F^2}{k_0^2}} \quad (3)$$

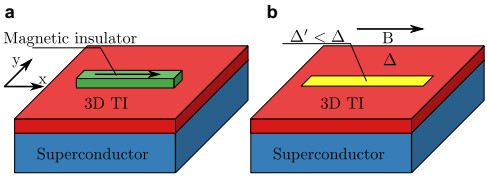

**Fig. 1 Platforms for creating Majorana bound states from segmented Fermi surface.** Here the examples are based on proximitized 3D topological insulator (TI), such as Bi$_2$Se$_3$, with superconducting gap $\Delta$. Quasi-1D channel is formed due to: **a** narrow strip of magnetic insulator (such as EuS) with magnetization parallel to the strip axis, **b** narrow region (yellow) with a smaller superconducting gap $\Delta'$ under external in-plane magnetic field $B$.

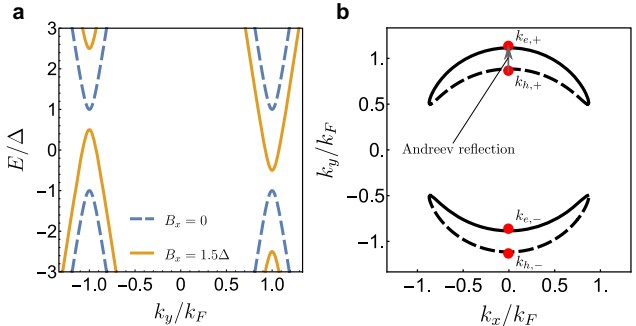

**Fig. 2 Segmented Fermi surface. a** Bands of the projected Hamiltonian (2) at wavevector component $k_x = 0$ for Zeeman field $B_x = 0$ (dashed line) and $B_x = 1.5\Delta$ (solid line). **b** Segmented Fermi surface at $E = 0$ for $B_x = 2\Delta$ with electron (solid line) and hole (dashed line) arcs. The arrow indicates the Andreev reflection process at $k_x = 0$.

with $k_0 = k_F\sqrt{1 - k_x^2/k_F^2}$ and $s = \pm 1$ for electron- and hole-dominated states.

Before proceeding, we discuss the experimental feasibility of our proposal. As shown by ARPES measurements, proximitized thin films of $Bi_2Se_3$[44] display a single Fermi surface due to the Dirac surface state, without any bulk bands at the Fermi level. The superconducting gap at zero magnetic field is hard with $\Delta \approx 0.5$ meV and spatially uniform, which demonstrates the high quality of the proximitized TI thin films. In a latest quasiparticle interference (QPI) experiment on proximitized $Bi_2Te_3$, the gapless superconducting state with segmented Fermi surface has been achieved under in-plane magnetic field above 20 mT[45]. Closing of the proximity gap at such a low field is enabled by the Doppler effect in Bogoliubov quasiparticle dispersion induced by the screening supercurrent in the parent superconductor[46]. The vector potential corresponding to this Doppler effect is equivalent to a giant Zeeman field on the order of ~ $B\lambda_L$, where $\lambda_L$ is London penetration depth. Most importantly, the QPI patterns display distinct features due to scattering between the hotspots of the segmented Fermi surface, unimpeded by disorder and any bulk bands. However, due to the strong hexagonal warping of the surface state in $Bi_2Te_3$ that can introduce additional complications to the quasiparticle spectrum, using $Bi_2Se_3$ with negligible warping may be preferable. Nonetheless, all of these observations suggest that the experimental realization of our proposal is feasible.

To open up the topological gap in a region with a segmented Fermi surface, we use the quantum confinement effects by limiting the transverse size of the region with the non-zero Zeeman energy. In such a case, by surrounding the magnetic strip by a fully gapped superconducting region with zero Zeeman energy we effectively form a 1D channel with the number of quasiparticle modes determined by the width of the strip. The use of the superconducting gap to confine Bogoliubov quasiparticles distinguishes our proposal from the previous schemes based on nanowires, which use vacuum to confine electrons[7,8]. The idea of creating 1D topological channels by confinement using superconducting gap has been explored in other setups[5,12,31].

**Topological phase diagram from scattering approach**. To investigate the topological properties of this system, we first focus on the quasiparticle spectrum in a strip of finite width $W$ in $y$ direction and infinite length in $x$ direction with no disorder inside of the wire. In such a case, $k_x$ remains a good quantum number and we can obtain the in-gap states spectrum by solving a scattering problem (as discussed in Supplementary Note 2) along $y$

axis with states under the magnetic strip normalized to carry a unit quasiparticle current. For energies $|\epsilon| < \Delta$ there are no propagating states outside of the magnetic strip region and so at the interfaces at $y = \pm W/2$ normal and Andreev reflection can occur with no transmission into the surrounding superconductor. Andreev reflection in proximitized surface states of 3D TI has been intensively studied experimentally[47–49]. At zero energy, under the magnetic insulator strip, we have electron- and hole-dominated states moving in positive and negative $y$ direction with $k_y$ given by Eq. (3).

As the gap in the continuum model first closes at $k_x = 0$ as Zeeman energy is increased, to determine the condition for gap inversion and thus establish the boundaries of the topological phases in the $B_x - W$ parameter space it is enough to consider the states with no longitudinal momentum. Since $k_x$ is conserved in the scattering processes, this means that for such states normal reflection at the strip boundaries is forbidden due to the spin-momentum locking of Dirac surface states (states on the opposite sides of the Fermi surface are orthogonal). Together with the unitarity condition for the scattering matrix this means that Andreev reflection at $k_x = 0$ and $\epsilon = 0$ can be characterized by a single phase $\phi_A$ acquired by the particles during the reflection. Therefore, the scattering matrix $S_{\pm W/2}$ at the two interfaces is given by:

$$S_{W/2} = \begin{pmatrix} 0 & e^{i\phi_A} \\ e^{-i\phi_A} & 0 \end{pmatrix}, \quad S_{-W/2} = S_{W/2}^* \quad (4)$$

Since the particles propagate freely between Andreev reflections at opposite interfaces, they acquire the phase determined by their wavevectors. This translates to transmission matrices for movement along the positive and negative $y$ directions $T_\pm = \text{diag}(\exp(ik_{e,\pm}W), \exp(ik_{h,\mp}W))$. To find the bound state spectrum we use the condition[50]:

$$\det\left(1 - T_- S_{W/2} T_+ S_{-W/2}\right) = 0 \quad (5)$$

Since we evaluate this condition for $\epsilon = 0$ and $k_x = 0$, it greatly simplifies to the form:

$$1 - \cos(2\Delta k W - 2\phi_A) = 0 \quad (6)$$

where $\Delta k = (k_{e,+} - k_{h,+})/2 = \sqrt{B_x^2 - \Delta^2}/v$ is the wavevector $y$ component difference between the electron and hole-like states. This allows us to derive the topological phase boundaries in the $B_x - W$ space to be:

$$W/\xi = \frac{\phi_A + \pi n}{\sqrt{\tilde{B}_x^2 - 1}}, \quad n \in \mathbb{N} \quad (7)$$

with superconducting coherence length $\xi = v/\Delta$, $\tilde{B}_x = B_x/\Delta$ and Andreev reflection phase determined from the microscopic considerations:

$$\phi_A = \text{Arg}\left(\frac{i - \exp(-\text{arcosh}\tilde{B}_x)}{-i + \exp(\text{arcosh}\tilde{B}_x)}\right) \quad (8)$$

We can also determine the quasiparticle spectrum at $k_x = 0$ in the vicinity of the phase boundaries. To do this, we solve Eq. (5) for $\epsilon$. We expand the solution close to $\tilde{B}_{c,n}$, which is the Zeeman energy at which the gap at $k_x = 0$ closes at the $n$th boundary. In doing so we get:

$$\epsilon_\pm = \pm\Delta \frac{\tilde{B}_{c,n}^2 W + \xi}{\tilde{B}_{c,n}^2(W + \xi)}(\tilde{B}_x - \tilde{B}_{c,n}) \quad (9)$$

where $\pm$ gives the two particle-hole symmetric eigenvalues.

As pointed out by Kitaev[1], the topological phase of a 1D superconductor is determined by the gap closing at $k_x = 0$. This is exactly described by Eq. (7), as it determines the conditions for the subsequent quasiparticle branches to cross $\epsilon = 0$ and become inverted. Therefore, crossing the boundaries with even $n$ marks the transition from the trivial to the topological regime, and crossing odd $n$ curves marks the opposite transition. We will be focusing on the first topological region between the $n = 0$ and $n = 1$ boundaries.

It is worth highlighting the two key advantages of our proposed setup. First, the phase boundaries and the number of subgap quasiparticle modes are independent of the chemical potential and instead rely only on the $W/\xi$ and $B_x/\Delta$ ratios. Remarkably, this remains true even in the presence of Fermi wavelength mismatch inside and outside of the strip. This is a result of the spin-helical nature of the TI surface state that forbids normal reflection at $k_x = 0$, leaving Andreev reflection as the only confinement mechanism at $k_x = 0$. As a consequence, even for a large chemical potential there is only one subgap mode as long as $W \sim \xi$. This is in contrast to the semiconductor nanowires, which can have many low-energy subbands that greatly complicate the phase diagram, introducing numerous topological phase transitions. Secondly, since a large portion of the original Fermi surface remains gapped, there will be no detrimental low-energy electrons moving parallel to the channel. These states, unaffected by the normal and Andreev reflection processes, would be present if the narrow strip region was in normal state. With such states out of the picture, the maximum topological gap at given $B_x$ will be determined by the energy of states at $k_x = 0$. Therefore, the upper bound on the topological gap is given by the crossing points of the subsequent branches of Eq. (9). This constitutes a strong enhancement of the achievable topological gap when compared to the platforms based on Josephson junctions with normal state weak link. Both of the described features result from confining the quasiparticles with segmented Fermi surface by Andreev reflection, rather than electrons with full Fermi surface by normal reflection, to a 1D channel.

**Comparison with numerical simulation.** We can now compare the approximate analytical solutions to the numerical calculations based on a tight-binding model (see Methods section) with translational invariance in $x$ direction and periodic boundary conditions in $y$ direction. The simulations were performed using the Kwant code[51]. First, we illustrate the difference between our proposal and a scenario in which there is no superconducting pairing under the magnetic strip. In Fig. 3a we present the numerical spectrum of 1D subbands without and with superconductivity, with the same potential barrier placed along the strip at its center to introduce normal reflection. In both cases we observe inverted quasiparticle branches inside of the superconducting gap. However, when superconductivity is absent in the weak link, there are low energy states with large $k_x$ (moving parallel to the strip) that will interfere with the observation of Majorana modes. On the contrary, with superconductivity present under the magnetic strip, there are no low energy states at large $k_x$ and the topological gap is several times larger with the same barrier strength. We can thus further investigate the $k_x = 0$ eigenvalues numerically to characterize the upper bound on the topological gap. We present these eigenvalues in Fig. 3b, which shows the subsequent branch crossings. We note that the formula of Eq. (9) provides a very good approximation of the energies for given $n$ up until it crosses with the $n + 1$ branch. We therefore highlight that the maximum upper bound is given by such crossing energies and this value can be larger than 60% of the original superconducting gap, which constitutes a significant

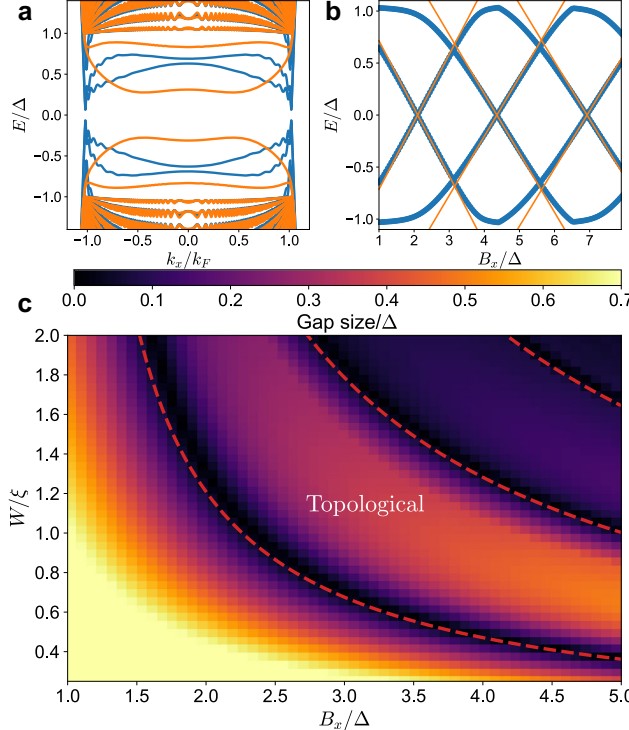

**Fig. 3 Spectrum of quasi-1D strip. a** Subgap states in a narrow strip at Zeeman field $B_x = 2.6\Delta$ and strip width $W/\xi = 1.16$ without (blue line) and with (orange line) superconductivity under the magnetic strip. **b** Bound state energies at $k_x = 0$ for increasing Zeeman energy. Solid orange lines indicate subsequent analytical solutions of Eq. (9). **c** The topological phase diagram of the system. Dashed lines indicate the phase boundaries described by Eq. (7).

advantage of our proposal. Finally, we investigate the full phase diagram of the system with potential barrier in Fig. 3c. The phase boundaries given by Eq. (7) are in very good agreement with the tight-binding calculation and the first topological region between curves $n = 0$ and $n = 1$ covers a wide area both in terms of the Zeeman energy $B_x$ and the strip width $W$, avoiding the necessity of parameter fine-tuning. We also note that the topological gap is a significant fraction of the original superconducting gap in the majority of the first topological region. In each following region the gap becomes smaller, but as we want to minimize the Zeeman energy necessary to obtain the topological phase this is not a concern. In general, the wider the strip, the smaller the Zeeman energy required to invert the branches. However, at the same time the possible size of the topological gap at the same normal reflection barrier strength decreases. Therefore, for the purpose of the experiment it will be necessary to find a sweet spot for a particular realization that maximizes the gap, based on a more precise modeling.

To further verify the topological character of the system, we perform numerical simulations of a finite length strip to demonstrate that it hosts Majorana bound states at its ends. First, we perform diagonalization of the 2D tight-binding Hamiltonian and plot the obtained eigenvalues in the form of density of states of the system in Fig. 4a. We observe, in accordance with the analysis of the phase diagram, that for Zeeman energies below ~2.1$\Delta$ the system is gapped and there are no zero energy states. However, at $B_{c,0} \approx 2.1\Delta$ the gap closes and then reopens due to normal reflection processes with a pair of zero energy states present in the system. The topological gap increases to above $0.25\Delta$ for this particular scattering strength

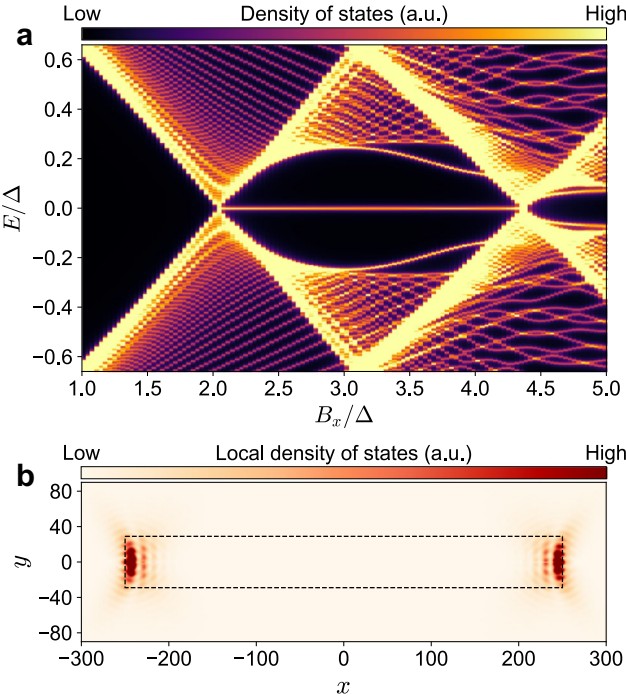

**Fig. 4 Finite length strip characterization. a** Density of states of the system with strip width $W/\xi = 1.16$. After the first gap closing a pair of zero energy states appears. **b** Local density of states at $E = 0$ for Zeeman field $B_x = 2.6\Delta$. The wave functions are strongly localized at the ends of the strip.

(not saturating the upper bound of $k_x = 0$) and then closes again at $B_{c,1}$ when the second pair of subbands is inverted. The gap reopens again, but this time we enter the trivial regime with no zero energy bound states as anticipated from the phase diagram. Finally, we plot the pair of zero energy states at $B_x = 2.6\Delta$, which are strongly localized at the ends of the strip as expected for the Majorana bound states.

## Discussion

In the preceding sections we have shown that using a narrow strip of a magnetic insulator, such as EuS, on top of a 3D topological insulator in proximity to a conventional superconductor can yield Majorana bound states with a large topological gap. However, our approach is not limited to Zeeman field induced by an adjacent magnetic insulator. Alternatively, if one applies external in-plane magnetic field, the same type of gap inversion will occur due to the effect of screening supercurrent in the parent superconductor. If the proximity-induced superconducting gap is non-uniform, e.g. with a narrow region of diminished magnitude $\Delta' < \Delta$, as shown in Fig. 1b, then the gap inversion will occur in that area, but not in the surrounding superconductor, effectively realizing the scenario we discussed in our work. One possible way of achieving this situation is to use the terraces of varying thickness naturally present in thin films of 3D topological insulators. As the strength of proximity effect that induces superconductivity in the surface state is dependent on the distance between the parent superconductor and the top surface[52], areas with thicker film will have reduced superconducting gap as compared to surrounding region with thinner film and larger gap. Moreover, as the external field scenario is based on the screening supercurrent modifying the quasiparticle dispersion via the Doppler effect, using thick (preferably bulk) superconductor is necessary for the supercurrent to flow without restrictions. In combination with using a material with long London penetration depth $\lambda_L$ (as the Cooper pair momentum $q \sim B\lambda_L$) this will lead to maximizing the effect.

Another possible system are the superconductors with Rashba spin-orbit coupling that have two independent pockets of spin-helical electrons (such as proximitized InAs 2D electron gas), resulting in two superimposed topological phase diagrams. However, due to the different velocities and superconducting gaps in each of the pockets, the superconducting coherence lengths will be different in each case, thus enabling the existence of a region with a single Majorana at each end. Such a scenario may help to understand the topological phase diagram of proximitized Rashba states in gold nanowires found in a recent study[28,31]. This largely expands the spectrum of material platforms for realization of our proposal, which is a testimony to its great versatility.

To sum up, in this work we proposed a new platform for creating Majorana bound states using proximitized Dirac surface states with in-plane Zeeman energy. Among the advantages of our proposal are: very small magnitude of required magnetic fields, removal of the spurious low energy states and large possible topological gaps. Together with the continuous progress in fabrication of thin 3D TI films coupled to superconductors this makes our proposal an attractive platform for studying Majorana bound states.

## Methods

**Tight-binding model for numerical calculations.** The numerical calculations were performed using a tight-binding approximation to the Hamiltonian (1), with terms included to ensure the presence of only a single Dirac cone in the Brillouin zone. The model is discretized on a square lattice with lattice constant $a = 1$, with the final Hamiltonian of the form:

$$H_{TB} = \sum_j c_j^\dagger (2t\tau_z s_z - \mu\tau_z + B_x(j_x, j_y)s_x + \Delta\tau_x)c_j +$$
$$\frac{t}{2}\left(c_{j+\hat{x}}^\dagger(-i\tau_z s_y - \tau_z s_z)c_j + c_{j+\hat{y}}^\dagger(i\tau_z s_x - \tau_z s_z)c_j + \text{H.c.}\right) \tag{10}$$

where $j = (j_x, j_y)$ is the index labeling each site of the tight-binding lattice, $\hat{x}$ and $\hat{y}$ are unit vectors of the lattice, $c_j$ is the vector of annihilation operators in Bogoliubov-de Gennes formalism at lattice site $j$, $t$ is the hopping strength between the neighboring sites, $\mu$ is the chemical potential, $\Delta$ is the superconducting gap parameter and $B_x(j_x, j_y)$ is the Zeeman energy, which is $B_x$ underneath the magnetic strip and 0 otherwise. In some of the calculations we also include a potential barrier along the magnetic strip at its center to introduce normal reflection into the system. The barrier Hamiltonian is given by:

$$H_{\text{barrier}} = \sum_{\substack{j:j_y=0, \\ -\frac{L_S}{2}<j_x<\frac{L_S}{2}}} V_b c_j^\dagger \tau_z c_j \tag{11}$$

In the calculations we use the following values of the parameters: $t = 1$, $\Delta = 0.02$. For the quasi-1D geometry we use $\mu = 0.8$ and $V_b = 0.8$, while for the finite length strip $\mu = 0.4$ and $V_b = 0.4$ to minimize the finite size effects. We change $B_x$ and the width of the strip $W$ as indicated for each simulation results figure.

We perform the simulations in two different configurations: (a) infinite strip with translational invariance in $x$ direction and (b) a finite strip of length $L_S$ that is fully surrounded by a gapped superconducting region of proximitized Dirac surface state. In both situations we apply periodic boundary conditions in $y$ direction. In the first case we keep the total system width $W_T = 300$ lattice sites. We can then calculate the spectrum as a function of the longitudinal momentum $k_x$ and in this way obtain the subgap quasiparticle modes as shown in Fig. 3a of the main text. This approach is also used to determine the phase diagram numerically, where the gap size plotted in Fig. 3c is obtained by finding the smallest positive eigenvalue over all of $k_x$. In the second case, we exactly diagonalize the full tight-binding Hamiltonian matrix with the total system size of width $W_T = 250$ and length $L_T = 500$ lattice sites. We then plot its eigenvalues as a function of the Zeeman energy in Fig. 4a. Figure 4b shows the local density of states at $E = 0$ obtained as $|\psi_1|^2 + |\psi_2|^2$, where $\psi_i$ are the two electron components of the wavefunctions obtained from the diagonalization procedure.

## Data availability
The data generated during the current study are available from authors upon reasonable request.

## Code availability
Code used to calculate the results presented in this work is available from the corresponding author upon a reasonable request.

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

## Acknowledgements

This work was supported by DOE Office of Basic Energy Sciences, Division of Materials Sciences and Engineering under Award DE-SC0019275. L.F. was supported in part by a Simons Investigator Award from the Simons Foundation.

## Author contributions

M.P. and L.F. have contributed in an essential way to all the aspects of this work.

## Competing interests

The authors declare no competing interests.
