## [Peer Review File · Nature Communications]

Editorial Note: Parts of this peer review file have been redacted as indicated to remove third-party material where no permission to publish could be obtained.

REVIEWER COMMENTS

Reviewer #1 (Remarks to the Author):

In this manuscript, the authors propose to use gapless superconductors with segmented Fermi surface to achieve Majorana bound state. In particular, they find that by placing a strip of magnetic insulator on top of a proximitized topological insulator thin film, Majorana zero modes will appear at the end of the strip. The authors also claim several advantages of their proposals over the existing Majorana platforms, which include the requirement of a small Zeeman field and the presence of a large topological gap. This theoretical proposal appears novel and the analysis is professional. However, most results in the current paper are built on a simplified model, which might have overlooked some realistic complications in experiments. As we have learned from other Majorana platforms (e.g. Rashba nanowires and FeTeSe), those realistic issues could lead to intrinsic difficulties in confirming the Majorana nature of an observed signal. Therefore, I am not fully convinced that this work will indeed inspire a new experimental Majorana platform and thus will be of interest to other experts in the Majorana community. There are several questions and concerns listed below, which should be addressed by the authors before any decision can be made.

(1) The authors have completely ignored the disorder effects throughout the discussion. For example, disorders might directly spoil the Majorana modes or introduce additional sub-gap bound state near zero energy, which, if happens, will weaken the claimed advantages for this platform. Therefore, it is important to study the robustness of the proposed Majorana physics under potential and magnetic disorders.

(2) The previous ARPES studies showed that the Fermi level of Bi₂Se₃ will cross some bulk bands. Since the Bi₂Se₃ thin film sits on a superconducting substrate and cannot be gated, I am wondering about how the existence of additional bulk bands will affect the current results. For example, can the bulk band introduce some subgap modes since it does not obey the spin-momentum locking?

(3) The authors should provide some estimates with realistic parameters for the proposed system with EuS on Bi₂Se₃, especially since both EuS and Bi₂Se₃ have been well-studied.

(4) The model for TI surface state only contains k-linear term. Since the authors have assumed a large chemical potential, the quadratic term and the cubic hexagonal warping term will enter the surface state Hamiltonian. I wonder how these higher-order corrections will modify the current results.

(5) While the authors have identified the gap-closing conditions and mapped out the phase diagram, they did not provide direct proof of why such gap closing must indicate a change of band topology. Thus, it will be necessary to calculate the Z₂ topological invariant for this effective 1d system, which will unambiguously clarify the topological nature.

(6) The authors claimed that the topological phase diagram is independent of the chemical potential. I don't think this is generally true and this feature is likely just an artifact of the over-simplified model. For example, if we push the chemical potential to well-above the band top of the conduction bulk band of TI, the Majorana physics will certainly disappear. I wonder if there will be a critical chemical potential, above which the 1d system is always topologically trivial. Understanding this critical chemical potential could enhance the tunability of the current setup, which has important consequences in possible braiding-related measurements (e.g. using T-junction geometry).

(7) Have the authors considered using FeTeSe as a platform to realize this proposal?

(8) I noticed a small energy splitting near zero energy in Fig. 4 (a), which occurs near $B_x \sim 3.7$. Is this a finite-size effect?

Reviewer #2 (Remarks to the Author):

The manuscript proposes a platform for realizing topological superconductivity and Majorana bound states that, in principle, presents some significant advantages over other platforms discussed in the literature (e.g., small Zeeman fields, no fine-tuning of the chemical potential, no detrimental low-energy states, and large topological gap). The platform consists of a 2D superconducting spin-helical system in the presence of in-plane Zeeman field, with a (topological) quasi-1D channel generated by quantum confinement of quasiparticles via Andreev reflection from a (fully gapped) surrounding superconductor. The work is motivated, in part, by the "multitude of issues plaguing current Majorana platforms." I find the proposal conceptually interesting, although not radically new (as it has significant similarities with other hybrid systems predicted to host Majorana bound states). Regarding the practical significance of the proposal, I am skeptical, as it is not obvious that the proposed platform will not suffer from some of the deficiencies "plaguing current Majorana platforms." A few specific points are highlighted below.

The schematic setup shown in Fig. 1 (together with Hamiltonian 1) suggest a TI film with gapless surface states on both the top and bottom surfaces. I would expect a large induced gap on the bottom surface (the TI-superconductor interface) and a much smaller (perhaps negligible) gap on the top ("active") surface. What indications do we have that the superconducting gap Δ (for the top surface) can be made large enough within this scheme?

The requirement of small Zeeman fields (to enter the topological regime) is mentioned as a potential advantage of this scheme. Fig. 3 suggests a typical Zeeman field on the order of 3Δ . This is not particularly small (as compared with, e.g., "standard" requirements for semiconductor-superconductor structures).

If proximity-induced, the Zeeman field is not tunable. This would leave W (the width of the strip) as the only parameter experimentally available for realizing the topological phase. Considering the difficulty of predicting theoretically the required value of W , building the proposed system would imply a significant amount of trial and error (or luck).

The chemical potential does not require fine tuning if it "naturally" resides within the bulk gap of the TI. It is not obvious that this is the case in a TI-superconductor hybrid structure. Moreover, if the chemical potential is not within the "right range", it would be extremely difficult (if possible at all) to control it.

The removal of spurious low energy states is listed as another potential advantage of this scheme. However, nothing is said about possible low-energy states induced by disorder and other inhomogeneities. It is worth remembering that other theoretically proposed schemes look quite promising at the level of idealized models. The problem stems from the imperfections that characterize real systems. There is no proof that such "imperfections" could not affect this structure. Moreover, considering the complexity of the structure (which will most likely involve significant material quality and device fabrication challenges), imperfections are expected to be present. It is not clear that the proposed scheme is more robust than other current platforms against "real life" issues.

What is the physical motivation for having a barrier in the middle of the strip? Is it associated with any expected feature of the (physical) system? What is the behavior of the model in the absence of a barrier?

Reply to Referees for “Creating Majorana modes from segmented Fermi surface”

Referee #1

In this manuscript, the authors propose to use gapless superconductors with segmented Fermi surface to achieve Majorana bound state. In particular, they find that by placing a strip of magnetic insulator on top of a proximitized topological insulator thin film, Majorana zero modes will appear at the end of the strip. The authors also claim several advantages of their proposals over the existing Majorana platforms, which include the requirement of a small Zeeman field and the presence of a large topological gap. This theoretical proposal appears novel and the analysis is professional. However, most results in the current paper are built on a simplified model, which might have overlooked some realistic complications in experiments. As we have learned from other Majorana platforms (e.g. Rashba nanowires and FeTeSe), those realistic issues could lead to intrinsic difficulties in confirming the Majorana nature of an observed signal. Therefore, I am not fully convinced that this work will indeed inspire a new experimental Majorana platform and thus will be of interest to other experts in the Majorana community. There are several questions and concerns listed below, which should be addressed by the authors before any decision can be made.

To begin with, we would like to thank the referee for their very thorough analysis of our manuscript and many insightful comments. Based on these comments, we have significantly expanded the discussion about the material considerations and the current experimental status. In particular, to answer the questions about the practical limitations of our proposal, we would like to draw referee’s attention to our recent collaborative work [arXiv:2010.02216] with experimental STM group that investigated the very system we are considering here. In that paper, quasiparticle interference (QPI) measurements were used to demonstrate the presence of segmented Fermi surface in thin films of Bi_2Te_3 grown on top of NbSe_2 . The QPI patterns display distinct features that can be directly attributed to scattering between the hotspots of the Bogoliubov quasiparticle Fermi surface. The details of this observation and how they resolve the issues raised by the referee are presented below.

However, we also stress that our work is significant from the conceptual perspective. The segmented Fermi surface can be created in many different ways (not limited to magnetic insulators, but also with external magnetic field), in multiple material platforms, such as Dirac surface states or semiconductors with strong spin-orbit coupling. We believe that the scenario in which the gapless superconducting state is obtained due to the influence of screening supercurrent in the parent superconductor is particularly interesting. Such a situation facilitates a significant enhancement of the magnetic field effect on the quasiparticle dispersion, which would be equivalent to a Zeeman effect with an enormous g-factor on the order of several hundred. Therefore, our work is setting up ground for further investigation of this phenomenon, both in the context of creating Majorana zero modes, as well as studies of finite momentum pairing and exotic superconducting states. That alone in our opinion gives our proposal a tremendous value.

(1) The authors have completely ignored the disorder effects throughout the discussion. For example, disorders might directly spoil the Majorana modes or introduce additional sub-gap bound state near zero energy, which, if happens, will weaken the claimed advantages for this platform. Therefore, it is important to study the robustness of the proposed Majorana physics under potential and magnetic disorders.

We thank the referee for this suggestion. As the referee pointed out, disorder is generally an important issue in topological superconductivity, and strong disorder is certainly detrimental to our proposal. However, a very recent STM experiment [arXiv:2010.02216] demonstrates that the thin films of 3D TIs in proximity with the superconductor NbSe₂ are of very high quality. The observed spectra are uniform over significant distances over 100 nm and the proximity gap induced in the surface state is large and hard. When an in-plane magnetic field is applied as we proposed, the quasiparticle interference pattern (QPI) is observed and displays distinct features that correspond directly to the scattering between the hotspots of segmented Fermi surface. Moreover, the observed features are strongly dependent on the orientation and strength of magnetic field, which is explained by the directional dependence of emergent Fermi surface pockets in the gapless superconducting state. If the disorder effect was disruptively strong, such QPI featuring clear k-space information would not be visible. This gives us the hope of practical implementation of our proposal. We have included discussion of the disorder impact in the revised manuscript.

The main focus of our work is a conceptually novel Majorana platform and analytical determination of its topological phase diagram in the clean limit. Again, we agree with the referee that disorder effect is important and plan to perform a detailed model study including disorder in a separate follow-up work.

(2) The previous ARPES studies showed that the Fermi level of Bi₂Se₃ will cross some bulk bands. Since the Bi₂Se₃ thin film sits on a superconducting substrate and cannot be gated, I am wondering about how the existence of additional bulk bands will affect the current results. For example, can the bulk band introduce some subgap modes since it does not obey the spin-momentum locking?

We thank the referee for raising this important point. To satisfy this concern, we would like to refer to an APRES measurement on thin films of Bi₂Se₃ [Nature Physics 10, 943–950 (2014)]. In that experiment, only the Dirac surface state is observed with a circular Fermi surface free from strong warping. Therefore, the bulk bands should not interfere with observation of Majorana zero modes in such a setup. Moreover, the experimental paper [arXiv:2010.02216] shows that quasiparticle interference pattern near the Fermi level is entirely due to Dirac surface states in proximitized Bi₂Te₃, as evidenced by the selection rule due to spin-momentum locking. No trace of bulk states was found in this STM measurement. Furthermore, even in the presence of bulk band, the proximity induced gap in such band should be larger than the one in the top Dirac surface state, so with a small magnetic field applied that gap will not close. This will significantly decrease the chance of additional low energy states appearing in the system. We have added a comment about the negligible impact of bulk bands to the main text.

(3) The authors should provide some estimates with realistic parameters for the proposed system with EuS on Bi₂Se₃, especially since both EuS and Bi₂Se₃ have been well-studied.

We agree that providing some estimates for realistic materials is important and we thank the referee for bringing that up. We have updated the manuscript with the parameters determined in the experiment using thin films of Bi₂Te₃ under in-plane magnetic field [arXiv:2010.02216]. In short, for a 4 quintuple layer thick film (4 nm) the proximity induced gap coming from parent superconductor NbSe₂ is about 0.5 meV. To close such a gap, magnetic field on the order of 20 mT is required. At the same time, such a magnetic field does not significantly impact the quasiparticle spectrum of NbSe₂ and no in-plane superconducting vortices were observed in the field of view. As this work concentrates on the conceptually novel approach to creating Majoranas using segmented Fermi surface, we leave specific material-oriented simulations for a future work.

(4) The model for TI surface state only contains k-linear term. Since the authors have assumed a large chemical potential, the quadratic term and the cubic hexagonal warping term will enter the surface state Hamiltonian. I wonder how these higher-order corrections will modify the current results.

First of all, the ARPES experiment on Bi₂Se₃ thin films [Nature Physics 10, 943–950 (2014)] shows that the Fermi surface of the surface states has a circular shape without any significant warping (compared to Bi₂Te₃). This justifies the usage of the linear Dirac model in our work. Moreover, the analytical calculation based on the k-linear terms serves the purpose of elucidating the physical mechanism for the appearance of Majorana zero modes due to the segmented Fermi surface. However, the formulas derived from the simplified model are then verified against numerical calculations based on a lattice model, which will necessarily include higher order terms. We have found a very good agreement between the analytical and numerical results, even for a large chemical potential in the simulation with Fermi surface occupying a large part of the Brillouin zone. To minimize the impact of hexagonal warping terms, one can use Bi₂Se₃ which has much weaker warping than Bi₂Te₃ and possess a circular Fermi surface. Nonetheless, even when hexagonal warping is present, the general idea of creating topological superconductivity from segmented Fermi surface will still apply.

(5) While the authors have identified the gap-closing conditions and mapped out the phase diagram, they did not provide direct proof of why such gap closing must indicate a change of band topology. Thus, it will be necessary to calculate the Z₂ topological invariant for this effective 1d system, which will unambiguously clarify the topological nature.

We thank the referee for an opportunity to reinforce this important point. As discussed by Kitaev, the topological phase of a 1D superconductor is determined by the difference of the parity of the spectrum at $k_x = 0$ and $k_x = \pi$. We have calculated the spectrum of quasi-1D system analytically, showing the system parameters for which the quasiparticle branches cross between 0 and π an odd number of times, indicating the change of the topological invariant. Moreover, we have explicitly calculated the spectrum for a finite length system, demonstrating the presence of a zero energy state that is localized at the 1D system boundaries, as expected for a Majorana zero mode in a 1D topological superconductor. We believe that this constitutes a sufficient proof of the topological nature of the system.

(6) The authors claimed that the topological phase diagram is independent of the chemical potential. I don't think this is generally true and this feature is likely just an artifact of the over-simplified model. For example, if we push the chemical potential to well-above the band top of the conduction bulk band of TI, the Majorana physics will certainly disappear. I wonder if there will be a critical chemical potential, above which the 1d system is always topologically trivial. Understanding this critical chemical potential could enhance the tunability of the current setup, which has important consequences in possible braiding-related measurements (e.g. using T-junction geometry).

We agree with the referee that in general the chemical potential is not infinitely tunable in our proposal. However, the assumption in our analytical calculation was that the chemical potential resides in the range in which the surface states provide a dominant contribution to the electronic behavior of the system and their dispersion still resembles a Dirac cone. When such assumptions are satisfied, the exact position of the Fermi level does not determine the number of 1D quasiparticle branches that enter the superconducting gap. This the advantage we have mentioned in the paper. As discussed above, the ARPES experiment performed on thin films of Bi₂Se₃ validates this assumption. Moreover, in the numerical

calculation we verified that results are largely independent of the Fermi energy and Majorana zero modes appear even for a Fermi surface that occupies a significant portion of the Brillouin zone.

(7) Have the authors considered using FeTeSe as a platform to realize this proposal?

Yes, we are investigating potential application of our proposal to the iron superconductor platforms. However, FeTeSe may not be best suited for this purpose due to its intrinsic inhomogeneity and strong disorder resulting from alloying. We are nonetheless hopeful that other iron superconductor compounds may alleviate these issues.

(8) I noticed a small energy splitting near zero energy in Fig. 4 (a), which occurs near $B_x \sim 3.7$. Is this a finite-size effect?

We thank the referee for noticing this issue. The energy splitting is indeed a result of finite size of the real space nanowire model. This has been improved in the new version of the manuscript with a revised Fig.4.

Referee #2

The manuscript proposes a platform for realizing topological superconductivity and Majorana bound states that, in principle, presents some significant advantages over other platforms discussed in the literature (e.g., small Zeeman fields, no fine-tuning of the chemical potential, no detrimental low-energy states, and large topological gap). The platform consists of a 2D superconducting spin-helical system in the presence of in-plane Zeeman field, with a (topological) quasi-1D channel generated by quantum confinement of quasiparticles via Andreev reflection from a (fully gapped) surrounding superconductor. The work is motivated, in part, by the “multitude of issues plaguing current Majorana platforms.” I find the proposal conceptually interesting, although not radically new (as it has significant similarities with other hybrid systems predicted to host Majorana bound states). Regarding the practical significance of the proposal, I am skeptical, as it is not obvious that the proposed platform will not suffer from some of the deficiencies “plaguing current Majorana platforms.” A few specific points are highlighted below.

We are grateful to Referee #2 for a deep and comprehensive analysis of our work and multiple interesting comments. These comments allowed to significantly improve the discussion of the experimental perspectives for our platform. Since the issues raised are pertaining to practical limitations of the proposal and are similar to those of Referee #1, we would also like to bring up the recent STM experiment [arXiv:2010.02216] of our collaborators. In that work, segmented Fermi surface has been observed in proximitized thin film of 3D topological insulator. This provides direct evidence that the foundation for our proposal is experimentally feasible and allows us to provide estimates for the expected system parameters. We provide more details based on these observations below specific comments. Nonetheless, we agree with the referee that some of the factors important in determining the advantages over current platforms require additional studies that we are planning to perform.

However, we also underline the conceptual novelty of our work. Our proposal in the most general sense can be realized using external magnetic field or magnetic insulators, in multiple different material platforms (including 3D TIs and semiconductors with strong spin-orbit coupling). We consider inducing topological phase transition using screening supercurrent in the base superconductor as particularly attractive. Such an approach can lead to a drastic decrease of the magnetic field required to close the gap, equivalent to having a huge g-factor on the order of several hundred. Thus, our work not only constitutes a new way of creating Majorana zero modes but is also setting up ground for further investigation of this

phenomenon in the context of finite momentum pairing and exotic superconducting states. That alone in our opinion gives our proposal a tremendous value.

The schematic setup shown in Fig. 1 (together with Hamiltonian 1) suggest a TI film with gapless surface states on both the top and bottom surfaces. I would expect a large induced gap on the bottom surface (the TI-superconductor interface) and a much smaller (perhaps negligible) gap on the top (“active”) surface. What indications do we have that the superconducting gap Δ (for the top surface) can be made large enough within this scheme?

We thank the referee for raising this important point as a large superconducting gap in the top surface state is crucial for clear observation of Majorana zero modes. Fortunately, this issue is clearly resolved by the STM experiment demonstrating the observation of segmented Fermi surface [arXiv:2010.02216]. First, the density of states measurements show that the superconducting gap on the top surface of 4 nm thick Bi_2Te_3 is about 0.5 meV. This is also consistent with the gap measured in the ARPES experiment in Bi_2Se_3 . Such a large gap is sufficient to resolve multiple in-gap features and holds promise for detailed study of Majorana physics. Moreover, the quasiparticle interference patterns unequivocally attribute this gap to a Dirac surface states. In previous QPI experiments on normal state of 3D TIs, the pattern consisted of 6 segments that correspond to the scattering between the tips of the hexagonally warped Fermi surface of Dirac fermions in Bi_2Te_3 . This observation has been repeated for energies far above the Fermi level in the proximitized thin films, together with no signal at the center of the gap. However, with in-plane magnetic field applied, the QPI pattern emerges also at zero energy, but is strikingly different from the normal state result. This time, the observed features depend on the direction of magnetic field and can be directly attributed to scattering between the pockets of segmented Fermi surface. We believe that these observations constitute sufficient proof that using the zero energy Fermi surface of Bogoliubov quasiparticles induced in the top surface of 3D TIs is experimentally feasible. We have included the information about the proximity-induced superconducting gap on the top surface of thin films in the new version of the paper.

The requirement of small Zeeman fields (to enter the topological regime) is mentioned as a potential advantage of this scheme. Fig. 3 suggests a typical Zeeman field on the order of 3Δ . This is not particularly small (as compared with, e.g., “standard” requirements for semiconductor-superconductor structures).

We agree that in normal circumstances, the Zeeman energy required for topological phase transition is large and for experimentally observed g-factors would translate to a magnetic field of 1 T. However, in addition to Zeeman coupling, an external in-plane magnetic field also induces a screening supercurrent in the parent superconductor, which introduces Doppler shift in the quasiparticle dispersion. The gap closing condition depends then on the Cooper pair momentum, which is determined by the penetration depth of the superconductor. This allows for a significant enhancement of the effective “Zeeman” energy to values comparable with the superconducting gap even for small magnetic fields. As shown in the experimental paper mentioned above [arXiv:2010.02216], the segmented Fermi surface arises above 20 mT, which in terms of the equivalent Zeeman energy would translate to an enormous g-factor on the order of several hundred. The experimental paper also shows that such a small field has a minimal impact on the parent superconductor, with a hard gap still present and no vortices within the field of view of the measurement. This enhancement due to the approach based on the screening supercurrent is why we believe that our

proposal improves upon the magnetic field requirements of the other models. We now comment on this issue in the main text.

If proximity-induced, the Zeeman field is not tunable. This would leave W (the width of the strip) as the only parameter experimentally available for realizing the topological phase. Considering the difficulty of predicting theoretically the required value of W , building the proposed system would imply a significant amount of trial and error (or luck).

We are grateful to the referee for pointing this out – the magnetic insulator based approach may indeed be more difficult to tune in a real experiment. However, the alternative scenario we present that uses in-plane magnetic field will circumvent the issue by allowing to tune the external magnetic field as well. Then it will be possible to adjust the position of the device on the phase diagram, limiting the importance of finding an optimal width of the strip by chance.

The chemical potential does not require fine tuning if it “naturally” resides within the bulk gap of the TI. It is not obvious that this is the case in a TI-superconductor hybrid structure. Moreover, if the chemical potential is not within the “right range”, it would be extremely difficult (if possible at all) to control it.

We certainly agree with the referee that tuning chemical potential is difficult in such hybrid structures. However, ARPES experiment on thin films of Bi_2Se_3 [Nature Physics 10, 943–950 (2014)] shows only a single Fermi surface of the topological surface state. This means that the bulk bands will not be contributing states that obscure observation of Majorana zero modes. The same conclusion has been reached from STM experiments studying the properties of proximitized thin film of Bi_2Te_3 . The reported quasiparticle interference patterns are strongly dominated by the surface state physics and the bulk bands contribution to the observed signal is almost non-existent. The reference to these experimental observations have been included in the revised manuscript.

This point raised by the referee shows exactly why our proposal is advantageous. As long as the bulk states do not obscure the physics of the system (and the two experiments discussed above demonstrate this), the exact position of chemical potential does not determine the number of in-gap quasiparticle branches that the topological phase of the superconductor depends on.

The removal of spurious low energy states is listed as another potential advantage of this scheme. However, nothing is said about possible low-energy states induced by disorder and other inhomogeneities. It is worth remembering that other theoretically proposed schemes look quite promising at the level of idealized models. The problem stems from the imperfections that characterize real systems. There is no proof that such “imperfections” could not affect this structure. Moreover, considering the complexity of the structure (which will most likely involve significant material quality and device fabrication challenges), imperfections are expected to be present. It is not clear that the proposed scheme is more robust than other current platforms against “real life” issues.

We definitely agree with the referee that considering disorder effects is important as they can be damaging to topological superconductivity. However, as discussed in response to Referee #1, the STM experiment we cite above demonstrates that the proximitized thin films of 3D TIs are already of very high quality, displaying remarkably rich details in their spectra. These details are in quantitative agreement with the model on which we base our proposal. The spectra themselves are largely unchanged when scanned across distances of over 100 nm and the proximity gap induced in the surface state is wide and

hard. Under an in-plane magnetic field the quasiparticle interference pattern (QPI) emerges at zero energy and displays distinct features that can be reliably explained by analyzing the hotspots of the segmented Fermi surface pockets. Furthermore, the observed patterns vary with the magnetic field direction and strength, which is also consistent with the behavior of the emergent segments of the Bogoliubov quasiparticle Fermi surface. If the disorder effect was disruptively strong, such QPI featuring clear k-space information would not be visible. This leads us to believe that our proposal is reasonable and may be soon realized experimentally. Nonetheless, we plan on further investigation of disorder effects in a separate work.

What is the physical motivation for having a barrier in the middle of the strip? Is it associated with any expected feature of the (physical) system? What is the behavior of the model in the absence of a barrier?

The presence of the barrier is associated with the unavoidable normal reflection that occurs for states with $k_x \neq 0$. In the idealized model without any disorder there is initially no source of additional scattering in the system and thus the gap opening will be resulting purely from quantum confinement effects (present independent of other circumstances). However, we wanted to test whether the proposed phenomenon will still be present when a strong source of normal reflection is included. In a real system the sources of normal reflection can include local variation in electrostatic potential in the confined region, either due to the presence of the magnetic insulator or some forms of gating of the surface. Normal reflection will also occur at the interface with the surrounding fully gapped superconductor.

List of changes to the manuscript:

1. Largely expanded discussion of the practical limitations of our proposal, with additional information about the current experimental status:
 - The absence of bulk states in thin films of 3D TIs as evidenced by ARPES and STM measurements
 - High quality of thin films with minimal impact of disorder that allows for resolving momentum space information
 - Presence of strong proximity effect in the top surface state with hard and wide superconducting gap
 - Source of enhanced effect of magnetic field on quasiparticle dispersion due to screening supercurrent
2. Added new references that support the claims about experimental realization of our platform
3. Improved Fig.4 which displays Majorana zero modes without any finite size effects that lead to splitting of the zero modes

REVIEWER COMMENTS

Reviewer #1 (Remarks to the Author):

My previous concern about this theoretical proposal is mainly about the experimental applicability. This is because many sample-dependent factors (especially the disorder effect) could spoil the validity of this proposal and further weaken the claimed advantages over other Majorana platforms. In their reply and revised manuscript, the authors have successfully addressed most of my previous concerns regarding the experimental feasibility. What really convinced me about the validity of this work is the follow-up experiment (arXiv:2010.02216), in which evidence of the proposed segmented Fermi surface has been observed. While there is still a long way to go before marching to experimentally achieve the Majorana proposal, the accompanied experimental efforts have made this proposal much more compelling and promising. Now I think this work will create a fairly large impact on the community, especially because of the supporting evidence observed in the experiment. In this regard, I would like to recommend this work for publication in Nature Communications.

Reviewer #2 (Remarks to the Author):

The authors have satisfactorily addressed many of the issues raised in the first round. However, most of their arguments rely heavily on i) the recent experimental results reported in arXiv:2010.02216 and ii) emphasizing the effect of screening supercurrents in the parent superconductor. Regarding the experimental results, they are encouraging, but far from demonstrating the feasibility of the proposal (as they only address the formation of a segmented Fermi surface, not the realization of the quasi-1D system that would support Majorana modes). Fig. 1(D) (from arXiv:2010.02216) shows that the Fermi level is within the bulk band, while Fig. 1(E) suggests the presence of a large (but relatively soft) induced gap. If this is the case, I expect the (top) "surface" states to be highly delocalized and the hexagonal warping to be significant. An explicit calculation that accounts for these effects may not be necessary, but including some comments explaining why the current formalism generates predictions that hold (qualitatively) in this regime would be helpful. Regarding point (ii), I believe that, given the potential significance of screening supercurrents, it is necessary to:

- a) Provide a brief explanation of this mechanism. In particular, why does this mechanism play a significant role in this proposal, while having negligible effects in other similar setups? What are the key parameters that control the strength of the effect due to screening supercurrents?
- b) How can one realize a quasi-1D region with a reduced induced gap [see Fig. 1(b)]? Again, the significance of this work will ultimately depend on "details" associated with its implementation. I definitely do not expect the authors to clarify all these details, but I think that it is important to provide enough elements that describe possible roadmaps for the implementation of the proposal and emphasize the (qualitative) differences with respect to the currently explored paths.

Reply to Referees for “Creating Majorana modes from segmented Fermi surface”

Referee #1

My previous concern about this theoretical proposal is mainly about the experimental applicability. This is because many sample-dependent factors (especially the disorder effect) could spoil the validity of this proposal and further weaken the claimed advantages over other Majorana platforms. In their reply and revised manuscript, the authors have successfully addressed most of my previous concerns regarding the experimental feasibility. What really convinced me about the validity of this work is the follow-up experiment (arXiv:2010.02216), in which evidence of the proposed segmented Fermi surface has been observed. While there is still a long way to go before marching to experimentally achieve the Majorana proposal, the accompanied experimental efforts have made this proposal much more compelling and promising. Now I think this work will create a fairly large impact on the community, especially because of the supporting evidence observed in the experiment. In this regard, I would like to recommend this work for publication in Nature Communications.

We thank the referee for recommending our article for publication. We completely agree that much follow up work is needed, both on the theoretical and experimental fronts, before the proposal can be reliably realized in the lab.

Referee #2

The authors have satisfactorily addressed many of the issues raised in the first round. However, most of their arguments rely heavily on i) the recent experimental results reported in arXiv:2010.02216 and ii) emphasizing the effect of screening supercurrents in the parent superconductor.

We are grateful to the referee for the second round of important suggestions that we used to further improve the clarity of exposition. Below we answer the remaining concerns point by point and highlight the changes in the manuscript that were made in order to satisfy the referee’s requests.

Regarding the experimental results, they are encouraging, but far from demonstrating the feasibility of the proposal (as they only address the formation of a segmented Fermi surface, not the realization of the quasi-1D system that would support Majorana modes). Fig. 1(D) (from arXiv:2010.02216) shows that the Fermi level is within the bulk band, while Fig. 1(E) suggests the presence of a large (but relatively soft) induced gap. If this is the case, I expect the (top) “surface” states to be highly delocalized and the hexagonal warping to be significant. An explicit calculation that accounts for these effects may not be necessary, but including some comments explaining why the current formalism generates predictions that hold (qualitatively) in this regime would be helpful.

We agree with the referee that the surface states in thin films of Bi_2Te_3 that were studied in the experiment presented in arXiv:2010.02216 exhibit strong hexagonal warping. While this was beneficial to the quasiparticle interference experiment as it introduced very distinct momentum space patterns due to the scattering between the hot spots of segmented Fermi surface, in the case of forming Majorana bound states it may have quantitatively important effects (although we believe it won’t change our phase diagram qualitatively). This is why we believe that for the purpose of realizing our model using Bi_2Se_3 is preferable. This material has much weaker hexagonal warping and ARPES measurements performed on thin films [Nature Physics 10, 943–950 (2014)] revealed the circular shape of the normal state Fermi surface, consistent with our analytical model. Moreover, these measurements also indicate the absence

of bulk conduction band states below 7 quintuple layers of thickness. As the apparent breadth of the superconducting coherence peaks in the proximitized thin film in arXiv:2010.02216 may be a result of superconductivity induced in the bulk conduction band, lack of such additional bulk states can improve the quality of the superconducting gap, increasing its hardness. Therefore, even though the general picture of formation of segmented Fermi surface should be valid irrespective of particular 3D TI used, we believe that using Bi_2Se_3 should be the optimal strategy to simplify the observed phenomena. For referee's convenience we attach below the relevant ARPES data from the Nature Physics paper cited above. We have expanded the discussion of the materials in the main text, highlighting the differences between Bi_2Te_3 and Bi_2Se_3 , especially in the context of hexagonal warping.

[Redacted]

Regarding point (ii), I believe that, given the potential significance of screening supercurrents, it is necessary to:

a) Provide a brief explanation of this mechanism. In particular, why does this mechanism play a significant role in this proposal, while having negligible effects in other similar setups? What are the key parameters that control the strength of the effect due to screening supercurrents?

We thank the referee for suggesting the expansion of the discussion of this mechanism. The key difference between our setup and the previous setups is the thickness of the superconductor used to proximitize the semiconductors. In our proposal, the superconductor is bulk and therefore the screening current can flow freely without spatial constraints, significantly increasing its effect on the quasiparticle spectrum via enhanced Cooper pair momentum. Moreover, the effect is dependent on the London penetration depth λ_L (as the Cooper pair momentum is $q \sim e B \lambda_L$). In the case of aluminum, this is about 16 nm, whereas in the case of NbSe_2 the penetration depth is over 100 nm, strongly enhancing the effect for the same external magnetic field. We have expanded the discussion on the screening supercurrent mechanism in the main text to clarify this issue.

b) How can one realize a quasi-1D region with a reduced induced gap [see Fig. 1(b)]? Again, the significance of this work will ultimately depend on "details" associated with its implementation. I definitely do not expect the authors to clarify all these details, but I think that it is important to provide enough elements

that describe possible roadmaps for the implementation of the proposal and emphasize the (qualitative) differences with respect to the currently explored paths.

One possible realization of the region with reduced proximity gap is the use of terraces naturally occurring in thin films of Bi_2Se_3 . As the strength of proximity effect in the surface state is dependent on the distance between it and the parent superconductor [Phys. Rev. Lett. 112, 217001 (2014)], each quintuple layer of thickness decreases the induced gap. Therefore, using a thicker region of the film surrounded by an area with a thinner film is an example of how to realize this scenario. However, since our proposal based on screening supercurrent induced topological phase transition is general both in terms of the source of superconducting gap closing and material platform, there are multiple setups in which this scenario can be realized. We certainly agree with the referee that investigating other possible setups in more detail is important in order to determine the optimal configuration in terms of materials and the device structure. We are leaving such analysis for future work. Nevertheless, to address this issue, we have included more details on this proposed setup in the main text.

List of changes to the manuscript:

1. Additional discussion of differences between Bi_2Te_3 and Bi_2Se_3 thin films regarding both the presence of bulk conduction band states and strength of hexagonal warping.
2. Expanded discussion of screening supercurrent-based mechanism.
3. More details on possible realizations of areas with reduced superconducting gap to implement our proposal using external magnetic field.